# Mediating and Moderating Mechanisms in the Relationship Between Social Media Use and Adolescent Aggression: A Scoping Review of Quantitative Evidence

**DOI:** 10.3390/ejihpe15060098

**Published:** 2025-06-03

**Authors:** Georgios Giannakopoulos, Afroditi Prassou

**Affiliations:** 1Department of Child Psychiatry, School of Medicine, National and Kapodistrian University of Athens, 115 27 Athens, Greece; 24th Directorate of Secondary Education of Athens, 171 21 Athens, Greece; aprassou@sch.gr

**Keywords:** adolescent aggression, social media use, mediators, moderators

## Abstract

Adolescents’ pervasive use of social media has been increasingly linked to aggression, including cyberbullying and hostile online interactions. While this association is well documented, the psychological and contextual mechanisms that mediate or moderate it remain unclear. This scoping review maps quantitative evidence on mediators and moderators between social media use and aggression among adolescents. A comprehensive search using ProQuest’s Summon platform was conducted across PsycINFO, Scopus, PubMed, and Web of Science, following the PRISMA 2020 guidelines. Eligible studies, published between January 2020 and March 2025, included adolescents aged 11–18 and reported at least one statistical mediation or moderation analysis. Forty-four studies from 19 countries (N > 90,000) were thematically synthesized. Key mediators included problematic use, moral disengagement, depression, attention-seeking, and risky digital behaviors. Moderators included gender, body satisfaction, cultural setting, school type, and family attachment. Most of the studies used structural equation modeling or PROCESS macro, although cross-sectional designs predominated. Limitations included reliance on self-reports and inconsistent social media measures. The findings suggest that social media–aggression links are indirect and shaped by emotional, cognitive, and ecological factors. Multi-level interventions targeting digital literacy, moral reasoning, and resilience are needed. This review was not registered and received no external funding.

## 1. Introduction

In recent years, social media platforms such as Instagram, TikTok, and Snapchat have become central to adolescent life, shaping how youths communicate, present themselves, and navigate peer relationships ([12]; [27]). Although these platforms offer opportunities for self-expression and social connection, a growing body of research has documented their association with various forms of aggression, including cyberbullying, hostile messaging, and digital retaliation ([9]; [35]). Given that over 90% of adolescents in many countries use social media daily ([38]), understanding the psychosocial mechanisms linking digital engagement with aggression has become a public health priority.

While the correlation between social media use and adolescent aggression is well established ([24]), the recent scholarship has shifted toward identifying mediating and moderating variables that explain or condition this relationship. Mediators—such as problematic social media use, moral disengagement, or emotional vulnerabilities—shed light on how social media affects behavior ([14]; [23]; [41]). Moderators—including gender, age, cultural context, or body satisfaction—highlight the boundary conditions under which these effects vary ([13]; [17]; [44]). This focus aligns with ecological and developmental models that consider adolescent behavior as shaped by dynamic interactions between individual traits and social environments ([15]; [37]).

Several theoretical frameworks have guided this line of research. The General Aggression Model ([3]) posits that situational inputs (e.g., cybervictimization) influence aggressive outcomes via internal affective and cognitive processes. For example, [10] ([10]) found that cybervictimization predicted cyberbullying through a serial mediation involving problematic social media use and moral disengagement. Their cross-national study of Italian and Spanish preadolescents revealed consistency in these pathways, suggesting cultural invariance in the psychosocial processes that escalate online conflict into aggression.

Other frameworks, such as the SPIN model (Status Pursuit in Narcissism) and social learning theory, emphasize the role of self-enhancement motives and peer feedback loops. [44] ([44]) reported that narcissism predicted both cyberbullying and online prosocial behavior via attention-seeking, but not impulsivity, indicating strategic rather than reactive aggression. Loneliness moderated this relationship, amplifying the effects of narcissistic traits on online behavior. Similarly, [13] ([13]) found that social comparison orientation led to the perpetration of cyberbullying and victimization via envy, particularly among adolescents with low body satisfaction—a pattern highlighting the emotional costs of appearance-based comparisons on social networking sites.

The cognitive dimension of aggression has also been explored through moral disengagement and identity-based mechanisms. [37] ([37]) found that sadism predicted cyber aggression, and that moral identity moderated this relationship in complex ways—acting both as a buffer and, paradoxically, as a predictor of aggression depending on internalization. [47] ([47]) demonstrated that dark triad traits (Machiavellianism, narcissism, psychopathy) predicted cyber aggression via moral disengagement, with gender differences moderating these pathways.

At the contextual level, parental communication, school climate, and cultural norms shape digital behavior. [1] ([1]) reported that both parental and adolescent gender moderated the influence of perceived parental control on cyber behavior. [8] ([8]) found that internet self-efficacy and online social capital mediated the relationship between cyberbullying and psychological distress, but only in public school students—indicating that educational context can moderate the efficacy of protective mechanisms. [17] ([17]) showed that students of color progressed more robustly through the cognitive stages of bystander intervention, even though actual intervention behavior was more common among White students—demonstrating racial variation in how moral cognition translates to action.

Despite these insights, the literature remains fragmented. Most studies have investigated mediators or moderators in isolation, often using cross-sectional designs and homogeneous samples. No systematic or scoping review to date has comprehensively examined the psychological and contextual mechanisms that mediate or moderate the relationship between social media use and adolescent aggression. This review addresses that gap by mapping the breadth and characteristics of 44 peer-reviewed quantitative studies published between 2020 and 2025.

This mapping is guided by three research questions: (1) What are the most consistently identified mediators and moderators linking social media use to adolescent aggression? (2) How do these mechanisms vary across demographic, cultural, or educational groups? (3) What methodological approaches are used to test these mechanisms, and what are their strengths and limitations? By addressing these questions, this review offers a structured empirical foundation for future theory development, prevention strategies, and culturally responsive interventions targeting aggression in digital adolescence.

## 2. Scoping Review Methodology

This scoping review was conducted in accordance with the PRISMA extension (PRISMA-ScR) to the PRISMA 2020 guidelines ([29]; [42]), which provides guidance for mapping the scope and nature of existing evidence. The review aimed to identify emerging patterns and themes in quantitative studies on mediating and moderating mechanisms, rather than to aggregate effect sizes or assess the strength of evidence, as would be expected in a formal systematic review. As a scoping review, the purpose was to map the existing literature, rather than to assess the strength or direction of effects across studies, as would be expected in a systematic review or meta-analysis. The goal was not to evaluate the evidence base for causal inference but to chart the types of mediating and moderating mechanisms examined, the methods used to test them, and how these variables were conceptually and statistically operationalized. This review thus offers breadth and thematic structure, rather than exhaustive or quantitative synthesis.

No formal protocol was prepared for this review. However, all methodological decisions were determined in advance and are reported transparently to ensure reproducibility. No amendments were made, as the review was not registered and no protocol was prepared. Although the review adhered to transparent and reproducible methods, the absence of a pre-registered protocol further aligns it with a scoping review framework, which emphasizes exploratory mapping of a research field.

The literature search was performed in March 2025 using ProQuest’s Summon Discovery Service of the Library and Information Center at the National and Kapodistrian University of Athens. This platform aggregates results from a wide range of scholarly databases, including PsycINFO, Scopus, Web of Science, PubMed, and ScienceDirect, thus allowing for comprehensive coverage of interdisciplinary research in psychology, education, public health, and communication studies. The search covered studies published between January 2020 and March 2025.

The decision to restrict the review to studies published from 2020 onward was based on three key considerations: First, the digital media landscape has changed rapidly in recent years, with platforms such as TikTok and Instagram overtaking earlier networks in adolescent engagement patterns. Second, the period from 2020 to 2025 reflects a post-pandemic digital shift in youth behavior, marked by increased screen time, social isolation, and new forms of online interaction. Third, recent studies have increasingly adopted more sophisticated statistical models (e.g., PROCESS macro, structural equation modeling), allowing for a clearer and more theory-driven examination of mediating and moderating variables. This five-year window therefore captures the most methodologically and contextually relevant evidence available.

The following Boolean keyword combination was used: (“adolescent” OR “youth” OR “teenager”) AND (“social media” OR “social networking” OR “Instagram” OR “TikTok” OR “Facebook”) AND (“aggression” OR “cyberbullying” OR “hostility” OR “interpersonal violence”) AND (“mediator” OR “moderator” OR “mediation” OR “moderation” OR “path analysis” OR “structural equation modeling” OR “PROCESS macro”). Filters were applied to restrict the results to peer-reviewed journal articles written in English and available in full text.

The inclusion criteria required that the studies (a) were published in English in peer-reviewed journals between 2020 and 2025; (b) focused on adolescents, typically aged 11 to 18; (c) employed quantitative methodologies; (d) examined aggression or related behavioral outcomes in connection to social media use; and (e) statistically tested at least one mediating or moderating variable. Studies were excluded if they addressed child or adult samples exclusively, employed qualitative or mixed-methods designs without quantifiable mediation or moderation analysis, or did not assess aggression-related outcomes. The authors screened all records for eligibility, including title/abstract screening and full-text review. No automation tools were used.

After the initial screening of 87 titles and abstracts, 81 articles were retained for full-text review. Of these, 37 were excluded for various reasons, including the absence of statistical mediation or moderation testing, irrelevant outcome variables, qualitative design, or age groups outside the adolescent range. Ultimately, 44 studies met all of the inclusion criteria and were included in the synthesis. The selection process is illustrated in a PRISMA-compliant flow diagram (Figure 1).

The authors extracted data from all included studies using a standardized coding form. No automation tools were used, and no additional data were obtained from the study investigators. For each included study, the following variables were recorded: publication year, country of origin, sample size and characteristics, research design, type of aggression assessed (e.g., cyberbullying, digital dating abuse), social media constructs investigated (e.g., problematic use, exposure, communication behavior), mediators and moderators tested, statistical models employed (e.g., SEM, PROCESS macro), and key findings related to both direct and indirect effects.

### Analytical Framework

To ensure consistency in data synthesis, a structured analytical framework was applied. Each included study was reviewed for (a) type and definition of aggression outcome, (b) operationalization of social media use, (c) mediating and/or moderating variables tested, (d) statistical model employed (e.g., SEM, PROCESS macro), and (e) sample characteristics, including age, gender, and cultural context. Mediators and moderators were coded based on thematic categories derived from theory, as outlined below. Rather than evaluating effect sizes, the review focused on the presence, direction, and statistical significance of each mediational or moderational pathway. This framework enabled the identification of recurrent mechanisms and cross-study convergence.

Given the heterogeneity in study designs, measurement instruments, and conceptualizations of both social media use and aggression, a meta-analytic synthesis was deemed inappropriate. Instead, a narrative thematic synthesis was employed to identify converging patterns and recurring mechanisms across cultural, methodological, and theoretical contexts. No sensitivity analyses were conducted, as the review was based on narrative synthesis rather than quantitative meta-analysis. The studies were grouped according to the primary type of mediating or moderating variable that they examined—such as psychological traits, cognitive mechanisms, emotional vulnerabilities, or sociocultural factors. Particular attention was given to methodological quality, statistical modeling approaches (e.g., serial and moderated mediation), and model fit indices, where reported. All reported quantitative outcomes related to adolescent aggression—such as cyberbullying perpetration, cybervictimization, digital dating abuse, and online hostility—were included. When multiple outcomes or timepoints were reported, the primary outcome, as defined by the study authors, was extracted. No data were excluded based on measurement instruments or timing. No data transformation, imputation, or conversion procedures were applied. All extracted information was used as reported in the primary studies. Given the narrative nature of the synthesis, missing or incomplete summary statistics were not a limiting factor. Additional data were extracted on study characteristics such as publication year, country of origin, participant age and sample size, study design, variables related to social media use and aggression, mediators and moderators, and analytic approach.

The categorization of mediators and moderators was informed by psychological theory, particularly the General Aggression Model ([3]), ecological systems theory ([6]), and contemporary models of adolescent socio-emotional development. Mediators were grouped based on their theoretical role in behavioral regulation (e.g., problematic use), cognitive processing (e.g., moral disengagement), emotional vulnerability (e.g., depression), self-concept (e.g., identity clarity), and social context (e.g., school climate). Moderators were organized according to their level of influence—individual (e.g., gender), interpersonal (e.g., parent–child dynamics), or sociocultural (e.g., national or educational context)—in line with ecological systems theory. This framework provided a theoretically grounded structure for the thematic synthesis.

Funding sources and conflicts of interest were not consistently reported and were therefore excluded from data extraction. When information was unclear or missing, the best available interpretation of the text was used; no study authors were contacted. Effect measures such as regression coefficients, odds ratios, or *p*-values were not systematically extracted, due to variability in reporting formats. Instead, the findings were synthesized based on the direction and presence of statistically significant mediation and moderation effects. Appendix A summarizes the 44 included studies, highlighting study characteristics, tested mechanisms, and key findings.

Study quality was assessed by the authors through a narrative appraisal using the Mixed-Methods Appraisal Tool (MMAT; [16]), which evaluates five key domains: representativeness of participants, appropriateness of measurements, completeness of outcome data, adjustment for confounders, and nonresponse bias. No automation tools were used. Studies meeting at least four of the five criteria were considered to be of moderate-to-high quality. The majority of the studies (27 out of 44) met all five MMAT criteria, reflecting high methodological rigor. Common strengths included representative sampling, use of validated measurement instruments, and complete outcome reporting. However, approximately one in four studies partially addressed confounder adjustment, and one in five provided unclear reporting on nonresponse bias. Despite these limitations, the widespread use of sophisticated statistical modeling (e.g., SEM, PROCESS macro) across studies reinforces the reliability of the reported mediation and moderation effects.

The MMAT framework captured the completeness of outcome reporting at the study level; however, no formal assessment of reporting bias across the synthesis (e.g., publication bias or selective outcome reporting) was conducted. Funnel plot analysis and statistical tests were not applicable due to the narrative nature of the synthesis. Similarly, no formal assessment of certainty in the body of evidence was performed. While study-level quality was evaluated using the MMAT, frameworks such as GRADE were not applied to assess the overall confidence in outcome-level conclusions. The full MMAT results are presented in Appendix A.

## 3. Results

This scoping review mapped findings from 44 peer-reviewed quantitative studies conducted between 2020 and 2025, comprising data from over 90,000 adolescents across 19 countries. Although the studies varied in their design, sample composition, and conceptual frameworks, there was a high degree of convergence regarding the mediating and moderating mechanisms linking social media use to adolescent aggression. Most of the studies employed cross-sectional designs (n = 42), while only two utilized longitudinal or experimental methodologies. Structural equation modeling and PROCESS-based mediation/moderation analyses were the most frequently used analytic strategies. Thematic synthesis yielded five primary categories of mediators and four thematic groups of moderators, as outlined below.

It is important to emphasize that this is a scoping review, not a systematic review or meta-analysis. Therefore, the findings are not reported as standardized effect sizes or pooled estimates. Instead, the studies are grouped thematically based on common mechanisms and pathways to identify what has been studied, how often, and with what methodological approaches. This allows us to describe the landscape of existing evidence and identify theoretical and empirical gaps, which is consistent with the aims of scoping reviews.

### 3.1. Mediating Variables

The following categories of mediators were derived from an iterative thematic analysis informed by theory. Behavioral, cognitive, emotional, identity-related, and contextual domains were identified based on their alignment with ecological and psychosocial models of adolescent development and aggression. These categories reflect not only the nature of the variables but also their proposed functional role in linking social media use to aggressive outcomes.

#### 3.1.1. Problematic Social Media Use

A consistent body of evidence identified problematic social media use as a robust mediator between digital engagement and aggression. Studies conceptualized this construct as a pattern of compulsive, emotionally dysregulated, or maladaptive engagement with social networking platforms. [9] ([9], [10]) demonstrated that cybervictimization increased the likelihood of cyberbullying perpetration through sequential mediation by problematic social media use and moral disengagement. Similarly, [32] ([32]) found that problematic use partially mediated the relationship between cybervictimization and psychosomatic complaints, accounting for approximately 12 percent of the total effect.

#### 3.1.2. Moral Disengagement

Moral disengagement emerged as a key cognitive mediator across multiple studies, particularly those conducted in Italy, China, and Malaysia. In the work of [10] ([10]) and [33] ([33]), moral disengagement was found to mediate the relationship between problematic media use and cyberbullying. [41] ([41]) employed a serial mediation model and demonstrated that false self-presentation led to increased moral disengagement, which, in turn, predicted cyber aggression and eventual social withdrawal. [47] ([47]) showed that moral disengagement mediated the link between dark triad personality traits and cyber aggression, with gender moderating these effects, particularly among female adolescents.

#### 3.1.3. Emotional and Psychological Vulnerability

Numerous studies highlighted emotional vulnerabilities—such as depression, anxiety, attention-seeking, and distress—as mediating mechanisms. [44] ([44]) reported that attention-seeking behavior mediated the association between narcissism and both cyberbullying and prosocial online behaviors. [23] ([23]) identified depression as a mediating factor between cybervictimization and non-suicidal self-injury. [20] ([20]) added that loneliness and peer exclusion mediated the link between social media conflict and cybervictimization.

In addition to emotional distress, sleep quality also appears to mediate the relationship between problematic social media use and aggression. [22] ([22]) demonstrated that social media addiction predicted nighttime social media use and poorer sleep quality, which, in turn, led to greater aggression among early adolescents, suggesting that disrupted sleep may act as both a consequence of digital overuse and a driver of maladaptive behavior.

In the Turkish context, [11] ([11]) identified hostility and depression as serial mediators between low family social support and problematic internet use. [14] ([14]) provided further nuance, showing that psychiatric symptoms like somatization, hostility, and paranoid ideation increased smartphone overuse, while fear of missing out was linked specifically to anxiety and male gender, but not to depression or hostility.

#### 3.1.4. Self-Concept, Identity, and Resilience

Several studies examined how self-related processes moderated or mediated the link between digital experiences and aggression. [18] ([18]) found that risky online communication behaviors mediated the relationship between dating app use and various forms of victimization. [19] ([19]) demonstrated that psychological distress, including anxiety and emotional exhaustion, mediated the path from cyberbullying to suicidal ideation. [30] ([30]) found that self-concept clarity and self-esteem jointly shaped adolescents’ emotional sensitivity to problematic social media use. [31] ([31]) showed that narcissism and self-control mediated the relationship between game addiction and aggression, with narcissism accounting for nearly half of the total effect. [5] ([5]) identified psychological resilience as a full mediator between internet addiction and anger control. [40] ([40]) added that sexting and body self-esteem mediated the link between internet addiction and sexual online victimization.

#### 3.1.5. Cognitive and Social Processes

Cognitive appraisal and social learning mechanisms also emerged as explanatory pathways. [13] ([13]) identified envy on social media as a mediator between social comparison and both cyberbullying perpetration and victimization. [8] ([8]) reported that internet self-efficacy and online social capital mediated the relationship between victimization and mental health, although the latter was significant only among public school students. [28] ([28]) demonstrated that parental communication and self-efficacy jointly mediated the effect of cybervictimization on depression. [15] ([15]) linked personality traits such as narcissism and disinhibition to sexting and online grooming victimization. [25] ([25]) highlighted prejudice as a mediator between perceived threat, school climate, and cyber aggression. [36] ([36]) connected stereotypical gender beliefs to digital dating abuse via cognitive scripts regarding relational norms. [43] ([43]) emphasized societal mattering as a mediator between attitudinal disengagement and delinquent intent.

#### 3.1.6. Environmental and Sociocultural Stressors

[21] ([21]) found that, in Thailand’s deep south, negative upbringing and exposure to violence predicted cyberbullying, with emotional traits like frustration and paranoia mediating the relationship. [26] ([26]) showed that time spent online during the COVID-19 pandemic mediated the effects of social media use on cyberbullying, with gender moderating this relationship. [45] ([45]) reported that internet exposure was a stronger predictor of aggression–victimization than family or peer attachment in Malaysia. [35] ([35]) found that sleep deprivation mediated the effect of weekday screen time on academic performance.

### 3.2. Moderating Variables

Moderators were organized into individual-level (e.g., gender, personality traits), interpersonal-level (e.g., parental relationships), and sociocultural-level (e.g., school type, national context) domains. This structure reflects an ecological framework emphasizing the multiple contexts that shape adolescent digital behavior.

#### 3.2.1. Gender and Age

Gender emerged as a frequently tested moderator. [1] ([1]) found that parental and adolescent gender both moderated the link between parental control and adolescent behavior. [27] ([27]) demonstrated that boys were more likely to cyberbully when controlling for problematic use. [2] ([2]) observed age and gender interactions in cybervictimization patterns, while [7] ([7]) found age-related variation in the impact of cybergossip. Moreover, [4] ([4]) reported that parental demographic factors, including gender and education level, moderated the effectiveness of different mediation strategies on adolescent cyberbullying and cybervictimization, with technical mediation showing stronger effects among more educated parents.

#### 3.2.2. Loneliness and Body Satisfaction

[44] ([44]) found that loneliness amplified the relationship between narcissism and attention-seeking, intensifying the pathway to aggression. [13] ([13]) showed that body satisfaction buffered the envy pathway between social comparison and cyber aggression.

#### 3.2.3. Cultural and Contextual Moderators

[10] ([10]) confirmed model invariance across Italy and Spain. [8] ([8]) found that the school type moderated the buffering effect of online social capital. [1] ([1]) noted cultural differences between Indian and Singaporean adolescents in cyber conduct. [17] ([17]) identified racial differences in bystander intervention behavior in U.S. students. [4] ([4]), focusing on adolescents in Saudi Arabia, found that parental mediation strategies varied in effectiveness depending on cultural norms and parental education, underscoring the significance of sociocultural context in shaping digital parenting outcomes.

#### 3.2.4. Psychological Traits and Parent–Child Relationships

[37] ([37]) showed that moral internalization moderated the effect of sadism on aggression. [34] ([34]) found that empathy attenuated the effect of hostile attribution bias on online trolling. [39] ([39]) revealed that parent–child closeness buffered the effect of cyber dating abuse on anxiety and depression. In a related study, [4] ([4]) found that parental mediation strategies—particularly technical and restrictive approaches—were more effective in reducing cyberbullying behaviors when combined with higher levels of parental education and digital awareness, highlighting the moderating role of family-level factors. [46] ([46]) highlighted that belief in negotiable fate and strong parental relationships jointly buffered the effect of psychological distress on cyber aggression during environmental crises.

## 4. Discussion

This scoping review mapped the findings from 44 peer-reviewed quantitative studies published between 2020 and 2025, with the aim of identifying mediating and moderating mechanisms in the relationship between social media use and adolescent aggression. The analysis revealed a multidimensional network of behavioral, cognitive, emotional, and contextual pathways that link digital engagement to aggressive outcomes. The results underscore that social media is not inherently conducive to aggression; rather, it is the convergence of individual vulnerabilities and environmental conditions that determines its psychological effects.

This review does not aim to aggregate or compare effect sizes across studies but, rather, to map how different psychological and contextual factors operate as mechanisms. Each mediating or moderating factor was analyzed in relation to its theoretical significance, frequency of appearance, and methodological treatment across studies. For example, emotional variables such as depression or attention-seeking were not only frequent but also embedded in multivariate models that link internal states to digital aggression. Likewise, cultural and environmental moderators, including school type and parental education, were often tested using subgroup or interaction analyses, reflecting increased attention to ecological specificity. Although most of the studies were cross-sectional, their cumulative design choices revealed a pattern of conceptual alignment with ecological and psychosocial frameworks.

The thematic convergence was strongest for mediators like problematic social media use and moral disengagement, but less consistency was observed in identity-related variables such as self-concept or narcissism. For example, some studies ([31]) framed narcissism as a vulnerability, while others ([44]) linked it to strategic social navigation. Similarly, contextual mediators such as online self-efficacy ([8]) were shown to be protective only under specific educational settings, highlighting the role of the institutional environment as both a moderator and mediator. These divergences underscore the complex and sometimes contradictory ways in which psychological traits operate in digital spaces.

Problematic social media use emerged as the most recurrent and consistent mediator, often serving as the initial link in a chain of adverse psychological outcomes. Studies from both Western and Asian contexts, including those by [9] ([9], [10]) and [32] ([32]), showed that compulsive or dysregulated media use not only contributed directly to cyber aggression but also initiated subsequent mediational processes involving moral disengagement and psychosomatic distress. These findings align closely with the General Aggression Model ([3]), which posits that situational inputs such as cybervictimization influence behavior through internal states, including affect and cognition. In this context, problematic social media use acts as both a behavioral expression of distress and a vehicle through which adolescents become more exposed to risk-laden peer interactions and harmful content.

Moral disengagement was also a salient mechanism, appearing frequently in studies conducted across culturally diverse settings. It consistently mediated the relationship between problematic media use and aggression, often functioning as a cognitive rationalization process. Studies by [33] ([33]), [41] ([41]), and [47] ([47]) emphasized how moral disengagement interacts with broader constructs such as self-presentation, emotional regulation, and personality traits. These findings suggest that cyber aggression is often not simply impulsive but may be framed by adolescents as morally excusable or socially justified, particularly in environments where accountability is attenuated.

In terms of emotional mechanisms, several pathways were identified that link psychological vulnerability to aggression through social media use. Attention-seeking ([44]), depression ([23]), and perceived social exclusion ([20]) were key mediators. Emotional dysregulation emerged as a recurring theme, particularly in the sequential pathways identified by [11] ([11]), where hostility and depression jointly mediated the effect of family support deficits on problematic internet use. In a related finding, [22] ([22]) demonstrated that sleep disturbance—mediated by nighttime social media use—served as a physiological pathway linking social media addiction to adolescent aggression, suggesting that behavioral dysregulation may be reinforced by impaired rest and recovery cycles. These findings reinforce developmental models of adolescence as a period of emotional volatility, and they provide empirical support for models that link social media engagement to emotional compensation strategies that may inadvertently increase the risk of aggression.

Beyond emotional factors, studies also illuminated the role of self-concept clarity, identity formation, and digital resilience. [30] ([30]) showed that fragmented self-concept increased vulnerability to negative online interactions, while [31] ([31]) highlighted the dual mediating role of narcissism and self-control in the link between gaming and aggression. Taken together, these findings suggest that digital aggression is embedded within larger processes of adolescent identity construction, especially in contexts where social validation and peer approval are increasingly pursued online.

The review also identified several sociocultural and environmental stressors that contribute to aggressive digital behavior. [21] ([21]) linked negative upbringing and exposure to violence in politically unstable regions to cyberbullying, with emotional traits mediating the pathway. [45] ([45]) highlighted the predominance of internet exposure over familial attachments in predicting cyber aggression among Malaysian youths, while [26] ([26]) showed that pandemic-related increases in screen time mediated the effects of social media and gaming use on online aggression. These findings underscore that digital aggression cannot be fully understood without considering structural inequalities, environmental adversity, and broader social disruptions.

Moderating variables further nuanced the interpretation of these mediational pathways. Gender consistently shaped the strength and direction of the associations, as shown by [1] ([1]) and [27] ([27]), although some studies challenged conventional gender patterns by highlighting interaction effects with personality traits ([37]). Other moderators included age, body satisfaction, and school type ([8]; [13]), as well as parent–child closeness ([39]), parental education level ([4]), and belief in negotiable fate ([46]). These moderators suggest that aggression outcomes are shaped not only by what adolescents experience but also by who they are, where they are, and how they interpret these experiences.

This review offers several key contributions. First, it establishes that the relationship between social media use and adolescent aggression is not direct but structured by a constellation of mediating and moderating factors. Second, it underscores the need for interventions that address not only behavioral regulation but also emotional, cognitive, and contextual dimensions. Programs that promote empathy, self-concept clarity, digital literacy, and moral reasoning may prove especially effective. Third, this review highlights the importance of tailoring prevention strategies to demographic and cultural contexts. For instance, fostering online social capital in under-resourced school settings or enhancing parental communication in collectivist cultures may yield distinct benefits.

Finally, these findings challenge the tendency to pathologize adolescent media use in blanket terms. Rather than viewing youths as passive recipients of digital harm, several studies in this review suggest that they are active agents navigating complex social ecosystems, sometimes with resilience and strategy. Future research should expand on this agency-oriented perspective by adopting longitudinal and mixed-methods designs, leveraging ecological models, and incorporating real-time behavioral metrics.

In conclusion, this review demonstrates that adolescent aggression in digital contexts emerges from the interactions between psychological vulnerabilities, cognitive frameworks, and environmental constraints. These insights call for multi-level, developmentally informed, and culturally sensitive approaches to prevention and intervention in an era where digital experiences increasingly shape adolescent development.

### Limitations

This scoping review has several limitations. First, the search was restricted to English-language, peer-reviewed journal articles, which may have excluded relevant studies published in other languages or grey literature sources. Second, most of the included studies employed cross-sectional designs and self-report measures, limiting causal inference and introducing potential biases such as shared method variance. Third, variability in the measurement of key constructs—such as aggression and problematic social media use—made direct comparisons between studies difficult. Fourth, the narrative synthesis approach did not include effect size extraction or formal assessment of publication bias. Lastly, although a structured coding process was used, the review was conducted by a single author team without independent double screening or interrater reliability checks, which may have introduced interpretive bias.

## 5. Conclusions

This scoping review synthesized findings from 44 quantitative studies examining the mediating and moderating mechanisms between social media use and adolescent aggression. Rather than a direct relationship, the evidence reveals that digital aggression is shaped by a complex web of behavioral, cognitive, emotional, and contextual factors. Problematic social media use and moral disengagement were the most consistently identified mediators, while moderators such as gender, school type, and family attachment influenced the strength and direction of the effects. These findings underscore the importance of multi-level, theory-informed interventions that enhance digital resilience and psychosocial skills. Future research should prioritize longitudinal and mixed-methods designs to capture developmental trajectories and contextual nuance.

## Figures and Tables

**Figure 1 ejihpe-15-00098-f001:**
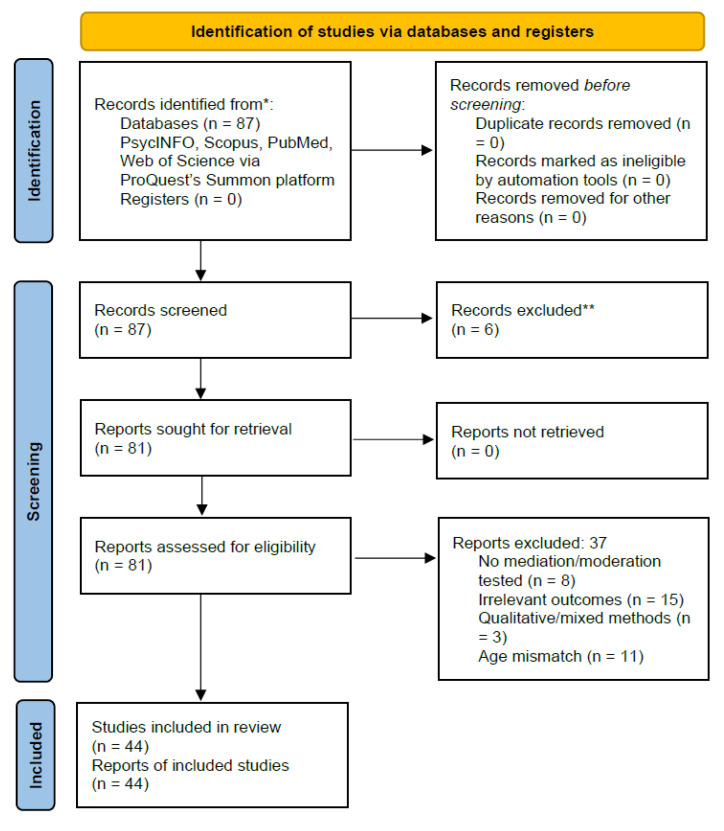
PRISMA flow diagram depicting the study selection process. * The total number across all databases/registers is reported. ** No automation tools were used for record exclusion.

## Data Availability

All data analyzed in this review were extracted from published studies and are summarized in the main manuscript and Appendix A. No new datasets were generated or analyzed during the current study. Additional information is available from the corresponding author upon reasonable request.

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
