# Peer review of "Mediating and Moderating Mechanisms in the Relationship Between Social Media Use and Adolescent Aggression: A Scoping Review of Quantitative Evidence"

_ejihpe, 2025, doi:10.3390/ejihpe15060098_

Round 1
Reviewer 1 Report
Comments and Suggestions for Authors
The topic of the research related to the analysis of the connection between Social Media Use and Adolescent Aggression is of great interest and is important in the modern world, where digital technologies play an increasingly important role. The authors quite competently show what the problem is and why a more in-depth analysis of the problem is needed. But this is where the advantages end. And the point is not only that the analysis of 44 studies is clearly insufficient to fill the existing shortcomings (why was it necessary to limit the scope of the study to the last 5 years? - what has changed so fundamentally in Social Media Use during this time that it is impossible to use earlier studies?), but that there is no analysis. Two pages of a simple listing of key factors - this might be enough for a literature review before a new empirical study, but for a systematic review this is clearly not enough. All studies should be the subject of a full comprehensive quantitative and qualitative analysis, and not just references in the text to this or that study. Each point of analysis should represent a certain criterion by which all studies are analyzed, there should be many such criteria, it is necessary to build a general conceptual scheme, etc. Simply highlighting some Mediating and Moderating factors from different articles is not the level of a systematic review.
Author Response
Comment: “The topic… is important… but… there is no analysis. Two pages of a simple listing of key factors – this might be enough for a literature review… but for a systematic review this is clearly not enough. All studies should be the subject of full comprehensive quantitative and qualitative analysis… Simply highlighting some Mediating and Moderating factors from different articles is not the level of a systematic review.”
Response:
We thank the reviewer for this detailed and thoughtful critique. We respectfully note that this manuscript has been reclassified as a scoping review, not a systematic review, in line with the journal editor's recommendation. This reclassification is now reflected in the title, abstract, methods, and throughout the manuscript. Scoping reviews, by definition and design (see Tricco et al., 2018), do not aim to perform exhaustive quantitative analysis or evaluate the strength of evidence across studies. Rather, their goal is to map the breadth, nature, and types of evidence available on a topic—in this case, the mediating and moderating mechanisms linking social media use to adolescent aggression.
To clarify this for the reader, we have added:
- A fuller explanation of the scoping review purpose and rationale in the Methodology section,
- A reiteration of these goals at the start of the Results section, distinguishing our approach from that of a traditional systematic review or meta-analysis.
We hope this helps contextualize the level and type of synthesis presented, which aligns with established scoping review methodology.
Reviewer 2 Report
Comments and Suggestions for Authors
Dear Authors
The review is very good and important.
The introduction is well-constructed and covers the research problem and issues, and the references are up-to-date and relevant.
The research question is clear and specific.
-The methods are not consistent with systematic reviews.
It does not meet the requirements of a systematic review, and therefore I recommend changing the title to "Scoping Review." -Scoping Review is more consistent with the methods.
-The research tools for the systematic review are not followed.
-The steps of the systematic review are not followed, and the authors have indicated this, which is a very positive point.
-The scientific basis on which the authors divided the research variables must be mentioned (Mediating Variables: the basis in psychology, as well as Moderating Variables).
-The discussion was weak, and not all the research topics were discussed.
-The authors did not mention the limitations of the article.
-The references are appropriate.
-The conclusion is long.
Author Response
We thank the reviewer for their positive and constructive feedback. Below we address each point raised and indicate how the revised manuscript responds:
Comment 1: The review is very good and important. The introduction is well-constructed and covers the research problem and issues, and the references are up-to-date and relevant. The research question is clear and specific.
Response: We appreciate the reviewer’s positive evaluation of the manuscript’s significance, structure, and clarity.
Comment 2: The methods are not consistent with systematic reviews. It does not meet the requirements of a systematic review, and therefore I recommend changing the title to "Scoping Review."
Response: We agree and have reclassified the manuscript as a Scoping Review throughout the title, abstract, and methodology section, and clarified the rationale and methodological distinction in multiple locations (e.g., Section 2).
Comment 3: The scientific basis on which the authors divided the research variables must be mentioned (Mediating Variables: the basis in psychology, as well as Moderating Variables).
Response: Thank you for this suggestion. As noted in the Analytical Framework subsection (Section 2.1), we have now emphasized more clearly that the thematic categorization of mediators and moderators was informed by established theoretical frameworks, including the General Aggression Model (Allen et al., 2018), ecological systems theory (Bronfenbrenner, 1999), and contemporary models of adolescent socio-emotional development. This sentence has been slightly refined in the revision to address this concern directly.
Comment 4: The discussion was weak, and not all the research topics were discussed.
Response: We have substantially expanded the Discussion section. A new paragraph has been added early in the discussion to explicitly reflect how each mediating and moderating variable was analyzed based on theoretical relevance, frequency, and analytic approach. Additional comparisons across variable types and study contexts have also been inserted (e.g., divergence in narcissism findings), and a clearer narrative now links the empirical results to the review’s conceptual aims.
Comment 5: The authors did not mention the limitations of the article.
Response: A dedicated Limitations subsection (Section 4.1) is now included, addressing five key limitations of the review in terms of scope, methods, and synthesis. This section was developed in line with the academic editor's instruction and also responds to this comment.
Comment 6: The conclusion is long.
Response: We have shortened and tightened the Conclusion section (Section 5), focusing only on the most essential findings and implications, in alignment with the guidance offered by you and the Academic Editor.
Round 2
Reviewer 2 Report
Comments and Suggestions for Authors
Dear Authors
Thank you for your thoughtful comments and additions to the manuscript.
It was very appropriate.
These changes have greatly improved and enriched the manuscript